# Determination of Triterpenoids and Phenolic Acids from *Sanguisorba officinalis* L. by HPLC-ELSD and Its Application

**DOI:** 10.3390/molecules26154505

**Published:** 2021-07-27

**Authors:** Jiahui Sun, Chunli Gan, Jing Huang, Zhenyue Wang, Chengcui Wu, Shuang Jiang, Xinrong Yang, Hesong Peng, Fanshu Wei, Chunjuan Yang

**Affiliations:** 1Department of Pharmaceutical Analysis and Analytical Chemistry, College of Pharmacy, Harbin Medical University, Harbin 150086, China; 2019020297@hrbmu.edu.cn (J.S.); 2020020323@hrbmu.edu.cn (C.W.); 2019020236@hrbmu.edu.cn (S.J.); 2019020237@hrbmu.edu.cn (X.Y.); 2020020324@hrbmu.edu.cn (H.P.); 2020020325@hrbmu.edu.cn (F.W.); 2Department of Medicinal Chemistry and Natural Medicine Chemistry, College of Pharmacy, Harbin Medical University, Harbin 150086, China; 101130@hrbmu.edu.cn; 3Department of Inorganic Chemistry, College of Pharmacy, Harbin Medical University, Harbin 150086, China; 101077@hrbmu.edu.cn; 4Department of Resources and Development of Chinese Materia Medica, College of Pharmacy, Heilongjiang University of Chinese Medicine, Harbin 150040, China; wangzhen_yue@163.com

**Keywords:** *Sanguisorba officinalis* L., phenolic acids, triterpenoids, HPLC-ELSD

## Abstract

A novel analytical method involving high-performance liquid chromatography with evaporative light scattering detection (HPLC-ELSD) was developed for simultaneous determination of 11 phenolic acids and 12 triterpenes in *Sanguisorba officinalis* L. Chromatographic separation was conducted with gradient elution mode by using a Diamonsil^TM^ C_18_ column (250 mm × 4.6 mm, 5 μm) with the mobile phase of 0.1% acetic acid water (A) and methanol (B). The drift tube temperature of ELSD was set at 70 °C and the nitrogen cumulative flow rate was 1.6 L/min. The method was fully validated to be linear over a wide concentration range (*R*^2^ ≥ 0.9991). The precisions (RSD) were less than 3.0% and the recoveries were between 97.7% and 101.4% for all compounds. The results indicated that this method is accurate and effective for the determination of 23 functional components in *Sanguisorba officinalis* L. and could also be successfully applied to study the influence of processing method on those functional components in *Sanguisorba officinalis* L.

## 1. Introduction

As a traditional Chinese medicine, *Sanguisorba officinalis L*. (*S. officinalis*) has a long history of medical application. It was first to be seen in Shennong Herbal Medicine Classic, the first great pharmacy book in China. The dried roots of *S. officinalis* (Rosaceae) and *Sanguisorba officinalis* var longifolia have been officially listed in the Chinese Pharmacopoeia (2020 edition) [1]. It has the functions of cooling blood, stopping bleeding, detoxifying, and curbing sores [2]. It has been widely used in the treatment of colitis, ulcerative colitis, pyelonephritis, hemostasis, and burns [3]. Besides, it was also reported to possess the following activities: antioxidant [4], anti-inflammatory [5], antiviral [6], antimicrobial [7,8], and antitumor [9]. Recently, more attention has been paid on its anti-cancer activities, such as antiproliferative effects against breast cancer [10], oral cancer [11], prostate cancer [12], and colorectal cancer [13].

The chemical constituents isolated from *S. officinalis* are mainly triterpene glycosides [14], phenolics, and flavonoids [15,16]. The polyphenols (Figure 1) include gallic acid (1), sanguiin H-4 (2), catechin (3), methyl-6-O-galloyl-*β*-D-glucopyranoside (4), caffeic acid (5), syringic acid (6), ethyl gallate (7), 3,4-dihydroxy-5-methoxybenzyl ester (8), ferulic acid (9), ellagic acid (10), and 3,3-dihydroxy ellagic acid (11). The triterpenes (Figure 1) include 2-oxo-3*β*,19*α*-dihydroxy-olean-12-en-28-oic acid *β*-D-glucopyranosyl ester (12), rosamultin (13), ziyuglycoside I (14), 3*β*-[(*α*-L-arabinopyranosyl)oxy]-urs-12,18(19)-dien-28-oic acid *β*-D-glucopyranosyl ester (15), 3*β*-[(*α*-L-arabinopyranosyl)oxy]-urs-12,19(29)-dien-28-oic acid *β*-D-glucopyranosyl ester (16), 3*β*,19*α*-dihydroxyurs-12-en-28-oic-acid 28-*β*-D-glucopyranosyl ester (17), pomonic acid (18), euscaphic acid (19), 2*α*, 3*α*-duhydroxyursa-12,19(29)-dien-28-oic acid *β*-D-glucopyranosyl ester (20), 1*β*-hydroxyeuscaphic acid (21), 3*β*,19*α*-dihydroxy-12-ursen-28-acid (22), and 2*α*-hydroxyoleanolic acid (23).

The Chinese Pharmacopoeia only selected gallic acid as a quality control indicator for *S. officinalis*, and there is a lack of simultaneous determination method for various triterpenes and phenolic acids. Using gallic acid as an indicator of quality control alone does not fully account for the quality of *S. officinalis* and the achievement of better control over its quality. These studies also show that the triterpenes in *S. officinalis* are more likely to be important components of its pharmacodynamic and physiological activity.

As these isolated components from *S. officinalis* have strong physiological activity [17,18,19], the determination of the content of these components in *S. officinalis* is therefore important, especially for the simultaneous determination of the content of the 23 compounds mentioned above. It is therefore essential to develop an analytical method that can determine the content of these 23 compounds simultaneously.

A HPLC-ELSD method was established to determine of triterpenoids and triterpenoid saponins in *Ilex pupurea* leaves [20]. A HPLC-DAD-ESI/MS(n) method was developed for quality evaluation of Cortex Moutan through identification of common constituents based on chromatographic fingerprints and determination of key pharmacological compounds [21]. However, it still cannot accurately evaluate the quality of other compounds. Although HPLC or gas chromatography (GC) coupled with mass spectrometry can quantify these compounds [22,23]. However, it has some practical limitations including cost and uncommon available in regular laboratories. Due to the terminal absorption of triterpenoids, UV as the analytical method has more interference. ELSD is commonly used in HPLC. The response of the ELSD does not depend on the optical properties of the substance under test and is not affected by its functional groups and can be detected for samples with volatility below the mobile phase. At the same time, the ELSD is insensitive to temperature changes, baseline stability, suitable for gradient elution of the liquid phase combination, more conducive to the separation of these compounds. Organic substances without ultraviolet absorption can be detected, such as triterpenes, ginsenosides, astragalosides, etc. [20]. Some reports have been successfully applied for the determination of oleanolic acid and ursolic acid in different medicinal plants by using HPLC-ELSD [24,25]. The general detection method of ELSD could eliminate the common difficulties in traditional methods [26].

For the reasons above, it is necessary to develop an analytical method to quantify the 23 functional ingredients to improve comprehensive exploitation and utilization of resources of *S. officinalis*. However, the method for the simultaneous determination of 23 compounds from *S. officinalis* with HPLC has not been reported. Thus, the aim of this study was to develop a simple, reliable method to simultaneously determine 23 compounds in *S. officinalis* by HPLC-ELSD. Meanwhile, it was the first time that the determination of 11 phenolic acids and 12 triterpenes together following *S. officinalis* extract was reported. This method could lay a good foundation for the comprehensive study of the active ingredients in *S. officinalis*, and also help to provide a reasonable and effective method for the quality evaluation of *S. officinalis*. In this study, the method was also used to study the influence of processing on the functional components in *Sanguisorba officinalis* L.

## 2. Results

### 2.1. Selection of HPLC-ELSD Condition

Determined the samples and 23 analytes according to the selected optimal chromatographic conditions. The representative HPLC-ELSD chromatograms of test samples and 23 analytes are shown in Figure 2.

According to the selected optimal chromatographic conditions, the retention times of 23 analytes are gallic acid (10.131 min), sanguiin H-4 (17.347 min), catechin (23.620 min), methyl-6-O-galloyl-β-D-glucopyranoside (26.492 min), caffeic acid (31.465 min), syringic acid (32.702 min), ethyl gallate (36.068 min), 3,4-dihydroxy-5-methoxybenzyl ester (39.079 min), ferulic acid (43.685 min), ellagic acid (51.505 min), 3,3-dihydroxy ellagic acid (57.338 min), 2-oxo-3β,19α-dihydroxy-olean-12-en-28-oic acid β-D-glucopyranosyl ester (66.723 min), rosamultin (73.744 min), ziyuglycoside I (78.160 min), 3β-[(α-L-arabinopyranosyl)oxy]-urs-12,18(19)-dien-28-oic acid β-D-glucopyranosyl ester (88.975 min), 3β-[(α-L-arabinopyranosyl)oxy]-urs-12,19(29)-dien-28-oic acid β-D-glucopyranosyl ester (94.293 min), 3β,19α-dihydroxyurs-12-en-28-oic-acid 28-β-D-glucopyranosyl ester (98.958 min), pomonic acid (101.456 min), euscaphic acid (103.725 min), 2α, 3α-duhydroxyursa-12,19(29)-dien-28-oic acid β-D-glucopyranosyl ester (105.710 min), 1β-hydroxyeuscaphic acid (108.168 min), 3β,19α-dihydroxy-12-ursen-28-acid (111.653 min), and 2α-hydroxyoleanolic acid (121.715 min).

### 2.2. Calibration Curves and Linear Range

The calibration curves were calculated by double logarithmic transformation of the area and the concentration of the reference solution injected. Linearity was evaluated by five different concentrations of standard solutions. For calibration, the standard solutions with five different concentrations were analyzed. A plot of peak area versus sample size by ELSD was not linear, but the plot of peak area versus sample size in double logarithm was linear, which conformed to the mechanism of the ELSD. Their regression equations were calculated in the form of *Y* =*a X* + *b*, where *Y* and *X* are the logarithms of the peak area and concentration, respectively. As shown in Table 1, compounds of different structures also show different slopes. The range and correlation coefficients are presented in Table 1. All calibration curves showed good linearity (*R*^2^ ≥ 0.9991) within the test ranges. Limits of quantitation ranged from 5.24 to 18.51 μg/mL. These results demonstrated that the HPLC-ELSD method is sensitive and precise for the quantitative determination of 23 analytes in *S. officinalis*.

### 2.3. Precision, Repeatablity and Stability

The experimental results of precision were 2.0%, 1.4%, 1.5%, 1.0%, 2.1%, 1.1%, 1.1%, 1.6%, 2.3%, 1.7%, 1.6%, 1.9%, 2.4%, 2.3%, 2.4%, 1.1%, 1.5%, 1.2%, 2.0%, 1.9%, 1.0%, 1.3%, 2.6%, 2.1%, respectively, which were all acceptable.

The result of repeatability shown as follows. The concentration was calculated by external standard method, and the RSD of 23 compounds were 1.5%, 1.3%, 2.1%, 0.49%, 0.56%, 1.4%, 0.48%, 1.7%, 0.30%, 1.1%, 1.5%, 1.4%, 2.0%, 0.47%, 0.78%, 0.19%, 0.36%, 1.4%, 0.52%, 2.4%, 0.29%, 0.41%, 2.0%, respectively, which met the requirement.

The RE among measurements ranged from −2.8% to 2.9% (Table 2.), which indicated that *S. officinalis* extract remained stable at least 12 h.

### 2.4. Recovery

The result was shown in Table 3. The recovery of 23 compounds ranged from 97.7% to 101.4% showed that the established method in this experiment has good accuracy.

### 2.5. Sample Analysis

Based on the experimental analysis above, the content of 11 phenolic acids and 12 triterpenoids in 32 batches of *S. officinalis* from 24 provinces was calculated. The source and collection time were shown in Table 4. The results of 23 compounds in 32 batches of *S. officinalis* were shown in Table 5 and Table 6. The distribution of the content of the 23 compounds in *S. officinalis* samples from different regions was listed in Figure 3.

All medicinal materials are purchased in the local Chinese medicine market in the 24 provinces mentioned below.

In Chinese Pharmacopoeia (2020 edition), gallic acid was used as a quantitative index in the medicinal materials and pieces of *S. officinalis*, and the content of gallic acid should not be less than 1.0%. From the experimental results, the content of gallic acid recorded in the Chinese Pharmacopoeia (2020 edition) was different in 32 batches of *S. officinalis*, but all the results met the quality control requirements recorded in the pharmacopoeia.

Under the current experimental conditions, in order to evaluate the quality of *S. officinalis* more reasonably and comprehensively, the content of phenolic acids and triterpenoids in *S. officinalis* was determined by HPLC. According to the content results, the contents of triterpenoids and phenolic acids in 32 batches are different, and some compounds cannot even be detected in *S. officinalis*. Among them, the content of ziyuglycoside I, catechin, and gallic acid in *S. officinalis* from Hebei was significantly higher than those in Gansu and Shanxi; the content of gallic acid and catechin in Shandong, Heilongjiang, and Shandong was higher; and the content of triterpenes in Inner Mongolia, Jiangxi, and Liaoning was also higher. The differences in the contents of triterpenoids and phenolic acids in *S. officinalis* are affected by many factors such as planting region, growing environment, and harvest time.

In this experiment, different batches of *S. officinalis* from different regions were tested. The results of the test are helpful to evaluate the quality and to control the internal quality of *S. officinalis*.

### 2.6. The Result of Comparison of Changes in Functional Components

The comparison of 23 compounds in unprocessed *S. officinalis* and processed *S. officinalis* are shown in Table 7 and Figure 4.

## 3. Discussion

The mobile phase combinations investigated included methanol–water, acetonitrile–water, methanol–0.1% acetic acid water, and acetonitrile–0.1% acetic acid water. The result showed that when methanol–0.1% acetic acid water was the mobile phase, 23 compounds in *S. officinalis* were well separated with symmetrical peaks and no tailing. The gradient elution method with different proportions was explored. The important parameter of ELSD conditions is the drift tube temperature which was evaluated at different drift tube temperatures from 60 °C to 80 °C. Good peak symmetries and efficiencies were achieved with a drift tube temperature at 70 °C. After examining a series of column temperatures, the final column temperature was determined to be 35 °C.

Linear of the method over a wide concentration range (*R*^2^ ≥ 0.9991). The precisions were less than 3.0% and the recoveries were between 97.7% and 101.4% for all compounds. The results indicated that this method is accurate and effective for the determination of 23 functional components in *Sanguisorba officinalis* L.

According to the result, compared with the unprocessed group, some compounds in *S. officinalis* after processed were significantly changed. The content of gallic acid, catechin, 3,4-dihydroxy-5-methoxybenzyl ester, ferulic acid, 3,3-dihydroxy ellagic acid, 3*β*-[(α-L-arabinopyranosyl)oxy]-urs-12,18(19)-dien-28-oic acid *β*-d-glucopyranosyl ester, 3*β*-[(α-L-arabinopyranosyl)oxy]-urs-12,19(29)-dien-28-oic acid *β*-D-glucopyranosyl ester, pomonic acid, 1*β*-hydroxyeuscaphic acid, and, 1*β*-hydroxyeuscaphic acid were decreased. Especially, compound (1), (3), (9), (11), (15), (16), and (21) were decreased significantly in batches from all three provinces. The content of methyl-6-O-galloyl-*β*-D-glucopyranoside, ethyl gallate, ellagic acid, and rosamultin was increased. Among them, (4) and (10) were increased significantly in batches from all three provinces. Sanguiin H-4 and 2*α*, 3*α*-dihydroxyursa-12,19(29)-dien-28-oic acid *β*-D-glucopyranosyl ester remain unchanged. Except for Heilongjiang, the content of caffeic acid in two batches of *S. officinalis* in Henan, Anhui decreased significantly. Syringic acid increased in batches from Heilongjiang, Anhui. 2-oxo-3,19-dihydroxy-olean-12-en-28-oic acid *β*-D-glucopyranosyl ester decreased in Heilongjiang and Anhui. Euscaphic acid and 3*β*,19*α*-dihydroxy-12-ursen-28-acid increased in Heilongjiang and Henan, but euscaphic acid decreased in batch from Anhui. In the batch from Heilongjiang, ziyuglycoside I and 3*β*,19*α*-dihydroxyurs-12-en-28-oic-acid 28-*β*-D-glucopyranosyl ester increased, but decreased in Henan and Anhui, respectively.

We processed the *S. officinalis* according to the method in the Chinese pharmacopoeia (the general rule 0213) and applied the established content determination method to determine the processed and unprocessed *S. officinalis* to compare the content changes of each compound. Which provides a theoretical basis for the different effects of unprocessed and processed *S. officinalis*. The changes in the content of these compounds may provide some ideas for the study of revealing the specific pharmacological effects of the active ingredients of *S. officinalis*.

## 4. Materials and Methods

### 4.1. Reagents and Samples

The purity of these 23 analytes refined in our laboratory (identified by NMR and MS) were all more than 98%. As reference standards, the NMR and MS data about the isolated compounds were shown in Table 8. The detailed identification results were published in the previous article of our laboratory [27,28]. Methanol and acetic acid (HPLC grade) were obtained from J&K Medical (Beijing, China). Ultra-pure water was extracted from a Milli-Q water purification system (Millipore, Molsheim, France). All other reagents were of analytical grade.

gallic acid (1): ESI-MS *m/z*: 171 [M+H]^+^, ^1^H-NMR (CD_3_OD, 400 MHz) *δ* 7.05 (2H, s, H-2, 6), ^13^C-NMR (CD_3_OD, 100 MHz) *δ* 121.6 (C-1), 109.9 (C-2,6), 146 (C-3), 139.2 (C-4), 170 (C-7).

sanguiin H-4 (2): ESI-MS *m/z*: 634 [M-H]^-^, ^1^H-NMR (CD_3_OD, 400 MHz) *δ* 4.19 (1H, m, H-1), 5.36 (1H, m, H-2), 5.95 (1H, m, H-3), 7.36 (1H, d, H-4), 3.60 (1H, m, H-6), 3.52 (2H, m, H-7), 6.77 (6H, s, H-16, 20, 25, 27, 35, 39), ^13^C-NMR (CD_3_OD, 100 MHz) *δ* 91.3 (C-1), 74.4 (C-2), 79.1 (C-3), 67.8 (C-4), 76.5 (C-5), 61.7 (C-6), 115.3 (C-1′), 126.3 (C-2′), 108.0 (C-3′), 145.8 (C-4′), 137.4 (C-5′), 146.0 (C-6′), 170.0 (C-7′), 115.4 (C-1″), 126.9 (C-2″), 107.6 (C-3″), 145.8 (C-4″), 137.5 (C-5″), 146.7 (C-6″), 171.4 (C-7″),*δ* 120.3 (C-1″‘), 110.5 (C-2″‘,6″‘), 146.7 (C-3″‘,5″‘), 140.6(C-4″‘), 166.4 (C-7″‘).

catechin (3): ESI-MS *m/z*: 291 [M+H]^+^, ^1^H-NMR (CD_3_OD, 400 MHz) *δ* 6.86 (1H, d, *J* = 2.0 Hz, H-2′), 6.78 (1H, d, *J* = 7.8 Hz, H-5′), 6.73 (1H, dd, *J* = 2.0, 7.8 Hz, H-6′), 5.95 (1H, d, *J* = 2.1 Hz, H-6), 5.88 (1H, d, *J* = 2.1 Hz, H-8), 4.58 (1H, d, *J* = 7.5 Hz, H-2), 3.98 (1H, *J* = 5.4, 7.5, 13.2 Hz, H-3), 2.85 (1H, dd, *J* = 5.4, 16.4 Hz, H-4a), 2.53 (1H, dd, *J* = 7.8, 16.4 Hz, H-4b), ^13^C-NMR (CD_3_OD, 100 MHz) *δ* 81.5 (C-2), 67.4 (C-3), 27.1 (C-4), 155.5 (C-5), 94.9 (C-6), 156.2 (C-7), 94.1 (C-8), 156.4 (C-9), 99.4 (C-10), 130.8 (C-1′), 113.9 (C-2′), 144.8 (C-3′), 144.8 (C-4′), 114.7 (C-5′), 118.7 (C-6′).

methyl-6-O-galloyl-*β*-D-glucopyranoside (4): ESI-MS *m/z*: 347 [M+H]^+^, ^1^H-NMR (Pyridine-*d*_5_, 400 MHz) *δ* 7.19 (2H, s, H-15,19), 4.96 (1H, dd, *J* = 1.6, 12.0 Hz, H-11a), 4.65 (1H, dd, *J* = 5.6, 12.0 Hz, H-11b), 4.18 (1H, d, *J* = 7.6 Hz, H-6), 3.98 (3H, s, H-21), ^13^C-NMR (Pyridine-*d_5_*, 100 MHz), *δ* 105.5 (C-1), 74.7 (C-2), 75.2 (C-3), 71.2 (C-4), 78.1 (C-5), 64.5(C-6), 56.4 (C-7), 167.1 (C-8), 121.0 (C-9), 110.0 (C-10, 14), 149.4 (C-11, 13), 135.2 (C-12).

3, 4-dihydroxy-5-methoxybenzyl ester (8): ESI-MS *m/z*: 197 [M-H]^-^, ^1^H-NMR (CD_3_OD, 400 MHz) *δ* 7.19 (1H, d, *J* = 1.96 Hz, H-2), 7.17 (1H, d, *J* = 1.96 Hz, H-6), 3.85 (3H, s, 3-OMe), 3.88 (3H, s, 7-OMe), ^13^C-NMR (CD_3_OD, 100 MHz) *δ* 121.5 (C-1), 106.1 (C-2), 149.2 (C-3), 140.6 (C-4), 146.3 (C-5), 111.9 (C-6), 168.9 (C-7), 52.4 (3-OMe), 56.7 (7-OMe).

rosamultin (13): ESI-MS *m/z*: 673 [M+Na]^+^, ^1^H-NMR (Pyridine-*d_5_*, 400 MHz) *δ* 6.30 (1H, d, *J* = 8.0 Hz, H-1″), 5.53 (1H, br.s, H-12), 3.36 (1H, d, *J* = 9.2 Hz, H-3), 2.92 (1H, s, H-18), 1.66 (3H, s, H-27), 1.38 (3H, s, H-29), 1.24 (3H, s, H-23). 1.20 (3H, s, H-26), 1.08 (3H, d, *J* = 6.4 Hz, H-30), 1.06 (3H, s, H-25), 1.05 (3H, s, H-24).

ziyuglycoside I (14): ESI-MS *m/z*: 789 [M+H]^+^, ^1^H-NMR (Pyridine-*d_5_*, 400 MHz) *δ* 6.34 (1H, d, *J* = 8.0 Hz, H-1″), 5.61 (1H, br.s, H-12), 4.82 (1H, d, *J* = 6.8 Hz, H-1′), 3.38 (1H, dd, *J* = 11.2, 3.2 Hz, H-3), 2.99 (1H, s, H-18), 1.75 (3H, s, H-27), 1.45 (3H, s, H-29), 1.32 (3H, s, H-23), 1.24 (3H, s, H-26), 1.12 (3H, d, *J* = 6.8 Hz, H-30), 1.02 (3H, s, H-24), 0.98 (3H, s, H-25).

3*β*-[(*α*-l-arabinopyranosyl)oxy]-urs-12,18(19)-dien-28-oic acid *β*-d-glucopyranosyl ester (15): ESI-MS *m/z*: 771 [M+Na]^+^, ^1^H-NMR (Pyridine-*d_5_*, 400 MHz) *δ* 6.33 (1H, d, *J* = 8.0 Hz, H-1″), 5.69 (1H, br.s, H-12), 4.77 (1H, d, *J* = 6.8 Hz, H-1′), 3.35 (1H, dd, *J* = 11.4, 4.2 Hz, H-3), 1.80 (3H, s, H-29). 1.36 (3H, s, H-23), 1.26 (3H, s, H-27), 1.12 (3H, d, *J* = 8.0 Hz, H-30), 1.03 (3H, s, H-26), 0.94 (3H, s, H-24), 0.88 (3H, s, H-25).

3*β*-[(*α*-l-arabinopyranosyl)oxy]-urs-12,19(29)-dien-28-oic acid *β*-D-glucopyranosyl ester (16): ESI-MS *m/z*: 771 [M+Na]^+^, ^1^H-NMR (Pyridine-*d_5_*, 400 MHz) *δ* 6.31 (1H, d, *J* = 8.1 Hz, H-1″), 5.53 (1H, br.s, H-12), 5.19 (1H, d, *J* = 1.3 Hz, H-29′), 5.04 (1H, d, *J* = 1.3 Hz, H-29″), 4.77 (1H, d, *J* = 6.8 Hz, H-1′), 3.83 (1H, s, H-18), 3.37 (1H, dd, *J* = 11.6, 4.1 Hz, H-3), 1.27 (3H, s, H-27), 1.27 (3H, s, H-23), 1.12 (3H, s, H-26), 1.05 (3H, d, *J* = 6.4 Hz, H-30), 0.95 (3H, s, H-24), 0.88 (3H, s, H-25).

3*β*,19*α*-dihydroxyurs-12-en-28-oic-acid 28-*β*-D-glucopyranosyl ester (17): ESI-MS *m/z*: 657 [M+Na]^+^, ^1^H-NMR (Pyridine-*d_5_*, 400 MHz) *δ* 6.34 (1H, d, *J* = 8.0 Hz, H-1″), 5.59 (1H, br.s, H-12), 3.45 (1H, dd, *J* = 9.9, 4.6 Hz, H-3), 2.96 (1H, s, H-18), 1.71 (3H, s, H-27), 1.43 (3H, s, H-29), 1.24 (3H, s, H-23), 1.24 (3H, s, H-26), 1.08 (3H, d, *J* = 6.6 Hz, H-30), 1.06 (3H, s, H-24), 0.98 (3H, s, H-25).

pomonic acid (18): ESI-MS *m/z*: 473 [M+H]^+^, ^1^H-NMR (Pyridine-*d_5_*, 400 MHz) *δ* 5.52 (1H, br. s, H-12), 4.35 (1H, m, H-3), 3.06 (1H, s, H-18), 1.73 (3H, s, H-27), 1.21 (3H, s, H-29), 1.13 (3H, s, H-23), 1.11 (3H, s, H-26), 1.06 (3H, d, *J* = 6.0 Hz, H-30), 1.00 (3H, s, H-24), 0.91 (3H, s, H-25).

2*α*, 3*α*-dihydroxyursa-12,19(29)-dien-28-oic acid *β*-D-glucopyranosyl ester (20): ESI-MS *m/z*: 655 [M+Na]^+^, ^1^H-NMR (Pyridine-*d_5_*, 400 MHz) *δ* 6.29 (1H, d, *J* = 7.6 Hz, H-1″), 5.50 (1H, br.s, H-12), 4.29 (1H, m, H-2), 3.78 (1H, br.s, H-3), 3.76 (1H, s, H-18), 1.25 (3H, s, H-23), 1.13 (3H, s, H-25), 1.11 (3H, s, H-27), 1.04(3H, d, *J* = 6.4 Hz, H-30), 0.98 (3H, s, H-26), 0.88 (3H, s, H-24).

1*β*-hydroxyeuscaphic acid (21): ESI-MS *m/z*: 527 [M+Na]^+^, ^1^H-NMR (Pyridine-*d_5_*, 400 MHz) *δ* 5.56 (1H, br.s, H-12), 4.13 (1H, d, *J* = 9.6 Hz, H-1), 4.16 (1H, dd, *J* = 2.5, 9.6 Hz, H-2), 3.86 (1H, d, *J* = 2.5 Hz, H-3), 3.03 (1H, s, H-18), 1.66 (3H, s, H-27), 1.41 (3H, s, H-29), 1.26 (3H, s, H-23), 1.25 (3H, s, H-25), 1.24 (3H, s, H-26), 1.11 (3H, d, *J* = 6.6 Hz, H-30), 0.92 (3H, s, H-24).

3*β*,19*α*-dihydroxy-12- ursen -28-acid (22): ESI-MS *m/z*: 473.4 [M+H]^+^, ^1^H-NMR (Pyridine-*d_5_*, 400 MHz) δ 5.61 (1H, br.s, H-12), 3.43 (1H, m, H-3), 3.05 (1H, s, H-18), 1.73 (3H, s, H-27), 1.23 (3H, s, H-29), 1.13 (3H, s, H-23), 1.11 (3H, s, H-26), 1.11 (3H, d, *J* = 6.0 Hz, H-30), 1.02 (3H, s, H-24), 0.91 (3H, s, H-25).

2*α*-hydroxyoleanolic acid (23): ESI-MS *m/z*: 456 [M]^+^, ^1^H-NMR (Pyridine-*d_5_*, 400 MHz) *δ* 5.45 (1H, t, *J* = 3.2 Hz, H-12), 3.37 (1H, dd, *J* = 5.8, 10.3 Hz, H-3), 3.27 (1H, dd, *J* = 4.1, 13.8 Hz, H-18), 1.26 (3H, s, H-23), 1.23 (3H, s, H-27), 1.06 (3H, s, H-29), 1.00 (3H, s, H-24), 0.97 (3H, s, H-30), 0.94 (3H, s, H-26), 0.93 (3H, s, H-25).

The LC/MS/MS data in the articles published in our laboratory was provided here for the discussion, as follows [29,30]: *m/z* 169.1→125.1 for gallic acid (1), *m/z* 633.1→300.9 for sanguiin H-4 (2), *m/z* 289.2→109.2 for catechin (3), *m/z* 179.0→135 for caffeic acid (5), *m/z* 197.0→122.8 for syringic acid (6), *m/z* 197.0→124.1 for ethyl gallate (7), *m/z* 197.1→182.2 for 3, 4-dihydroxy-5-methoxybenzyl ester (8), *m/z* 193.1→134 for ferulic acid (9), *m/z* 301→145 for ellagic acid (10), *m/z* 328.7→314 for 3, 3-dihydroxy ellagic acid (11), *m/z* 673.4→511.4 for rosamultin (13), *m/z* 784.5→437.4 for ziyuglycoside I (14), *m/z* 771.5→609.1 for 3*β*-[(*α*-L-arabinopyranosyl)oxy]-urs-12,18(19)-dien-28-oic acid *β*-D-glucopyranosyl ester (15), *m/z* 652.5→455.4 for 3*β*,19*α*-dihydroxyurs-12-en-28-oic-acid 28-*β*-D-glucopyranosyl ester (17), *m/z* 655.4→493.0 for 2*α*, 3*α*-dihydroxyursa-12,19(29)-dien-28-oic acid *β*-D-glucopyranosyl ester (20), *m/z* 505.2→423.2 for 1*β*-hydroxyeuscaphic acid (21). Through the LC/MS/MS data shows that the HPLC-ELSD method we established is useful and effective.

### 4.2. Preparation of S. officinalis Extract

After crushing the dried root of *S. officinalis* (1.0 g), it was extracted by hot reflux with 10 mL 70% ethanol solution (1:10, *w/v*) 3 times at 80 °C, 1 h each, and then filtrated. The combined filtrate was evaporated to steam, and the residue was dissolved in methanol to get a concentration equivalent to 0.005 g/mL of the *S. officinalis* extract.

### 4.3. Preparation of Standard Samples

Reference standards (1–23) were weighed accurately and dissolved by methanol in 10 mL volumetric bottle to yield a nominal concentration (205.2, 210.0, 601.2, 209.9, 300.1, 221.0, 210.4, 209.5, 420.1, 110.4, 210.2, 505.2, 901.0, 540.0, 470.0, 920.0, 940.0, 1060, 740.0, 800.0, 800.0, 450.0, 120.0 μg/mL, respectively). The concentrations of mixed reference solution of analytes (1–23) were 51.3, 52.5, 60.1, 52.5, 75.0, 55.3, 52.6, 52.4, 105.0, 55.2, 52.6, 126.3, 90.1, 135.0, 117.5, 92.0, 94.0, 106.0, 74.0, 80.0, 80.0, 45.0, 60.0 μg/mL, respectively. The working standard solutions were prepared from the concentrated stock solution by further appropriate dilution in the methanol. All the solutions were stored at 4 °C and filtered through the 0.45 μm nylon membrane prior to analysis.

### 4.4. Instrumentational and HPLC-ELSD Conditions

The analysis was performed on an Agilent series 1260 HPLC instrument coupled with an Agilent 1290 ELSD (Agilent Technologies, Santa Clara, CA, USA). The system was equipped with Diamonsil ^TM^ C_18_ column (250 mm × 4.6 mm, 5 μm) with the flow rate of 1.0 mL/min at 35 °C. The mobile phase was 0.1% acetic acid water (A) and methanol (B) with a gradient elution of 10–25% B at 0–20 min, 25–45% B at 20–50 min, 45–55% B at 50–55 min, 55–65% B at 55–70 min, 65–70% B at 70–90 min, 70–85% B at 90–115 min, 85–95% B at 115–125 min, 95–10% B at 125–130 min. The drift tube temperature of ELSD was 70 °C and nitrogen cumulative flow rate was 1.6 L/min.

### 4.5. Method Validation

#### 4.5.1. Calibration Curves, Limits of Detection and Quantification

For calibration, the premixed working solution with at least five different concentrations (performed in triplicate) was analyzed, and the linear calibration curves were calculated by plotting the logarithm of concentration injected for each analysis. The mixed solution of the 23 reference compounds was further diluted to a series of concentrations by methanol for the gain of limit of quantification (LOQ). The LOQ under the present chromatographic conditions were determined at the signal-to-noise ratios of each analyte of about 10, respectively.

#### 4.5.2. Precision, Repeatability and Stability

The precision of the quantitative method was investigated by determining the 23 analytes in six replicates within one day. The RSD value of the peak area was adopted to evaluate precision.

Repeatability was confirmed with six independent analytical sample solutions prepared according to the methods described above and variations expressed by RSD.

To assess the stability of the developed method, the sample was stored at room temperature and injected into the HPLC apparatus at 0, 2, 4, 6, 8, 12 h after preparation, respectively. Calculate the concentration of the 23 analytes at each time point and compare with the value of 0 h, to obtain RE (relative error).

#### 4.5.3. Recovery

Six parts of *S. officinalis* powder with known content, each of ~0.5 g, were weighted accurately, put in conical bottle. Moreover, reference substances (1–23) were added precisely. The test sample solutions were prepared as Section 4.2. Under the chromatographic conditions in Section 4.4., the peak area of each compound was recorded, and the recovery and RSD value of each compound were calculated. The recovery was expressed as: Recovery = (measured content—the known content)/added content of reference substances × 100%.

### 4.6. Content Determination

Thirty-two batches of *S. officinalis* powder from 24 provinces were weighed to a total of 1.0 g. The test sample solutions were prepared as the Section 4.2. The content of 23 compounds in *S. officinalis* was calculated with HPLC chromatographic conditions in Section 4.4.

### 4.7. Comparison of Changes in Functional Components in S. officinalis

*S. officinalis* powder from the same origin was divided into unprocessed group and processed group. The unprocessed group precisely weighed 1.0 g, *S. officinalis* of processed group was heated until the surface was black and the inside was brown, taken out and left to cool at room temperature, and precisely weighed to 1.0 g after processing. Both groups were prepared as Section 4.2. Measured under chromatographic conditions in Section 4.4. We then calculated the content of 23 compounds to compare the changes in functional ingredients.

## 5. Conclusions

In this study, a sensitive and reliable quantitative method for simultaneous analysis of phenolic acids and triterpenoids from *S. officinalis* was developed based on HPLC-ELSD. This is the first report of simultaneous determination of 11 phenolic acids and 12 triterpenoids in *S. officinalis* extract, so this method could be used to evaluate the quality of *S. officinalis*. At the same time, it provides a basis for the establishment of the characteristic and fingerprint of traditional Chinese medicine of *S. officinalis*. The method is of great significance for the quality control of *S. officinalis* resources and guiding its application.

Except for sanguiin H-4 and 2*α*, 3*α*-dihydroxyursa-12,19(29)-dien-28-oic acid *β*-D-glucopyranosyl ester, processed of *S. officinalis* has effect on all other compounds. Therefore, processed has a great impact on the 23 functional components of *S. officinalis*.

## Figures and Tables

**Figure 1 molecules-26-04505-f001:**
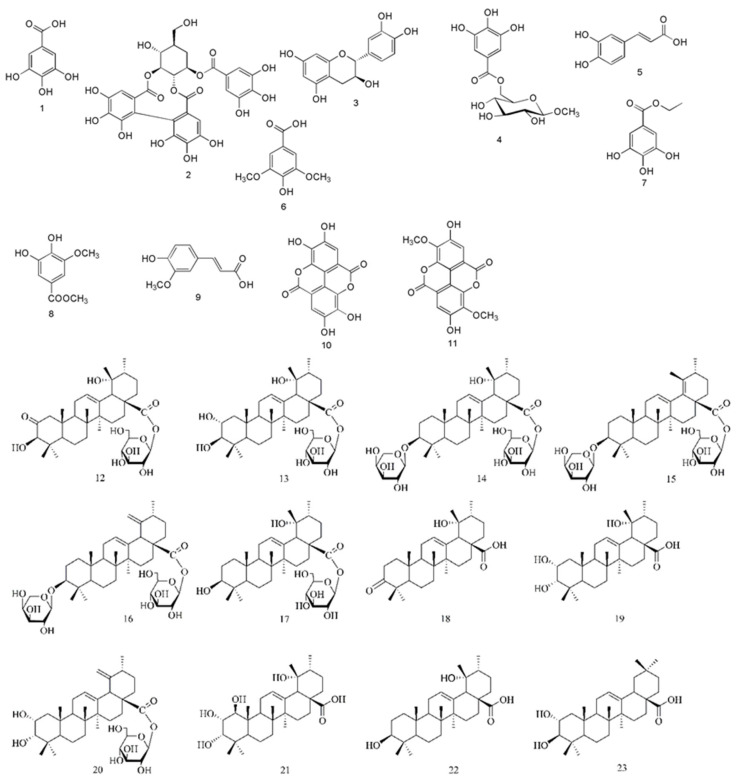
Chemical structures of 11 phenolic acids and 12 triterpenes.

**Figure 2 molecules-26-04505-f002:**
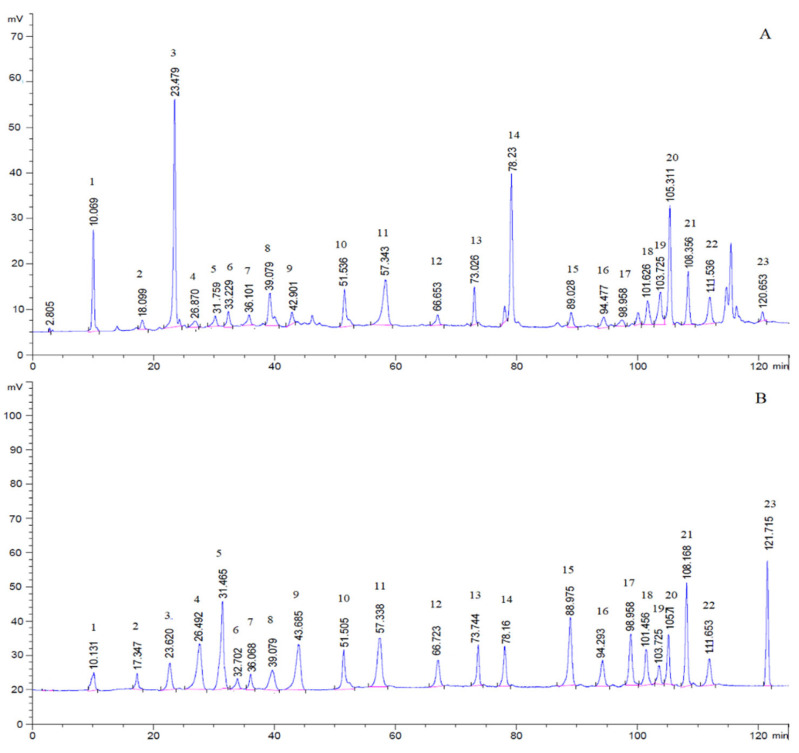
HPLC-ELSD chromatograms of test sample (**A**) and reference solutions (**B**): gallic acid (1), sanguiin H-4 (2), catechin (3), methyl-6-O-galloyl-*β*-D-glucopyranoside (4), caffeic acid (5), syringic acid (6), ethyl gallate (7), 3, 4-dihydroxy-5-methoxybenzyl ester (8), ferulic acid (9), ellagic acid (10), 3, 3-dihydroxy ellagic acid (11), 2-oxo-3*β*,19*α*-dihydroxy-olean-12-en-28-oic acid *β*-D-glucopyranosyl ester (12), rosamultin (13), ziyuglycoside I (14), 3*β*-[(*α*-L-arabinopyranosyl)oxy]-urs-12,18(19)-dien-28-oic acid *β*-D-glucopyranosyl ester (15), 3*β*-[(*α*-L-arabinopyranosyl)oxy]-urs-12,19(29)-dien-28-oic acid *β*-D-glucopyranosyl ester (16), 3*β*,19*α*-dihydroxyurs-12-en-28-oic-acid 28-*β*-D-glucopyranosyl ester (17), pomonic acid (18), euscaphic acid (19), 2*α*, 3*α*-dihydroxyursa-12,19(29)-dien-28-oic acid *β*-D-glucopyranosyl ester (20), 1*β*-hydroxyeuscaphic acid (21), 3*β*,19*α*-dihydroxy-12- ursen -28-acid (22), and 2*α*-hydroxyoleanolic acid (23).

**Figure 3 molecules-26-04505-f003:**
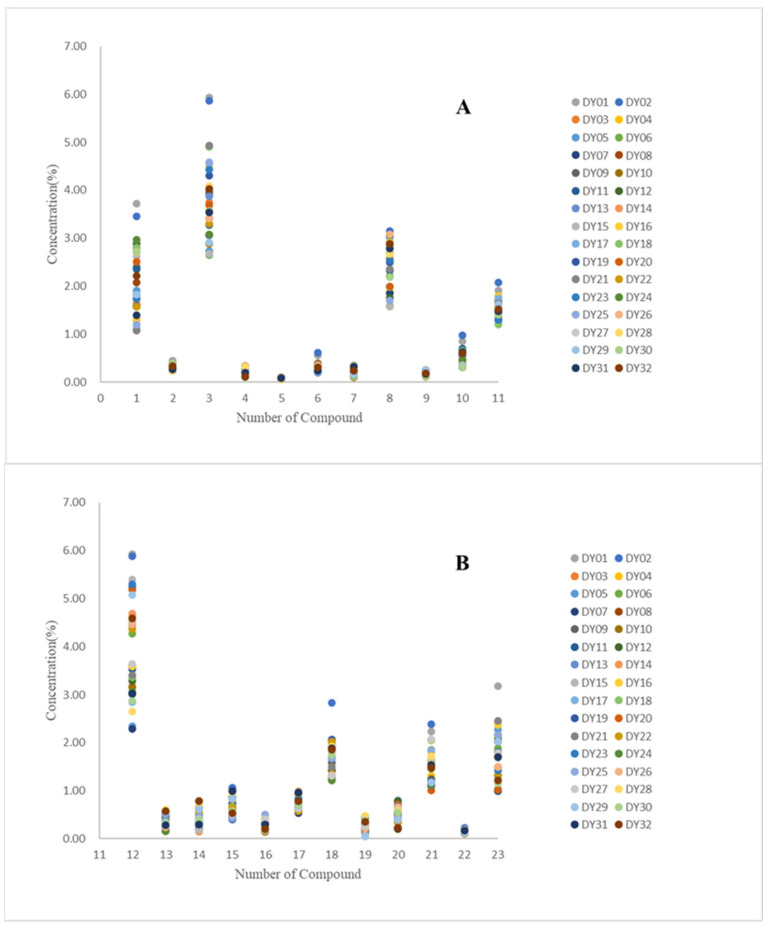
Distribution of the content of the 23 compounds in *S. officinalis* samples from different regions: (**A**) 11 phenolic acids and (**B**) 12 triterpenes.

**Figure 4 molecules-26-04505-f004:**
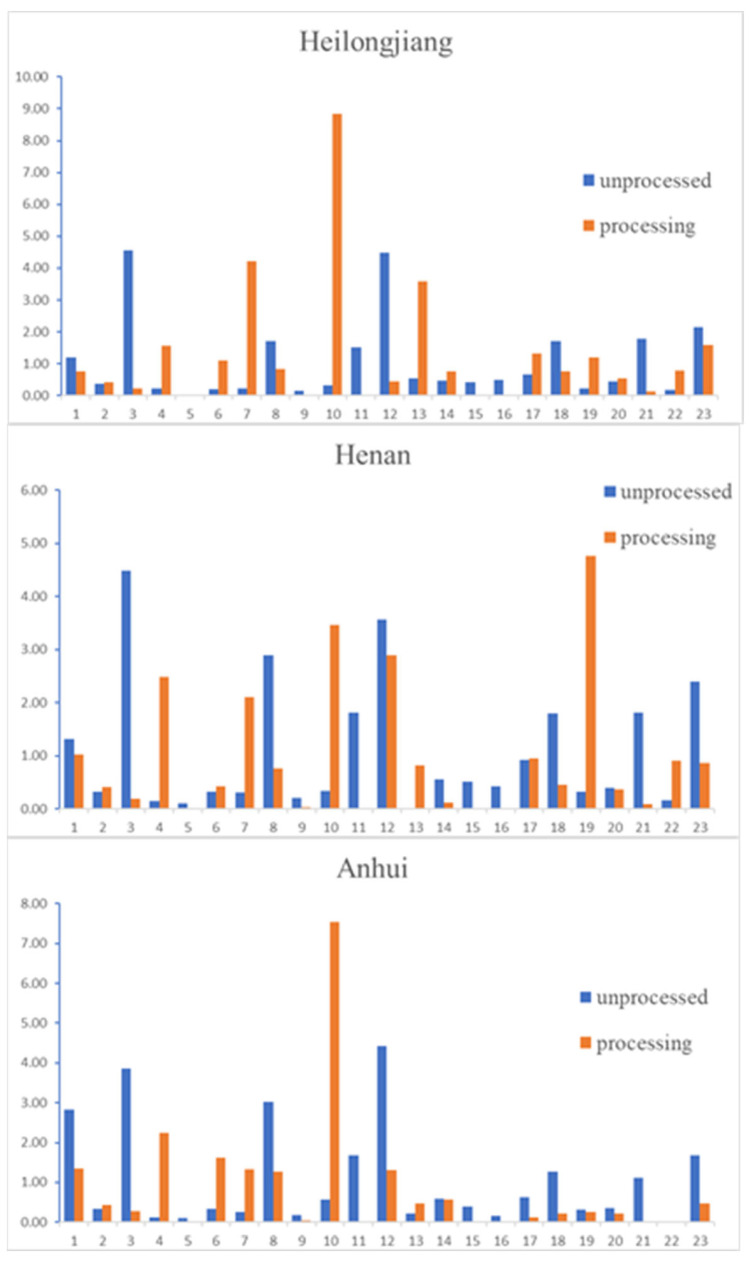
The content of the 23 compounds in unprocessed and processed *S. officinalis* samples from different regions.

**Table 1 molecules-26-04505-t001:** The calibrations and linear range of 23 analytes from *S. officinalis*.

Analyte	Calibration Curve	R²	Linear Range (μg/mL)
1	*Y* = 1.171*X* + 1.105	0.9993	10.25–164.0
2	*Y* = 1.114*X* + 1.482	0.9992	5.35–85.60
3	*Y* = 1.144*X* + 1.672	0.9996	7.88–126.0
4	*Y* = 1.028*X* + 0.6003	0.9992	7.55–120.8
5	*Y* = 0.8546*X* + 3.735	0.9996	8.08–209.2
6	*Y* = 1.162*X* + 1.196	0.9994	11.05–176.8
7	*Y* = 1.174*X* + 1.496	0.9992	5.45–87.20
8	*Y* = 1.156*X* + 1.109	0.9993	5.24–83.80
9	*Y* = 1.048*X* + 2.477	0.9992	10.50–168.0
10	*Y* = 1.092*X* + 2.293	0.9994	8.28–132.4
11	*Y* = 0.997*X* + 2.936	0.9994	10.10–160.6
12	*Y* = 1.126*X* + 1.429	0.9997	12.60–201.6
13	*Y* = 1.138*X* + 1.476	0.9991	11.80–188.8
14	*Y* = 1.131*X* + 1.629	0.9996	11.30–180.8
15	*Y* = 1.128*X* + 2.354	0.9997	11.35–181.6
16	*Y* = 1.135*X* + 2.403	0.9996	5.40–86.40
17	*Y* = 1.109*X* + 1.949	0.9996	11.55–184.8
18	*Y* = 1.141*X* + 2.017	0.9996	8.05–128.8
19	*Y* = 1.076*X* + 1.598	0.9994	8.10–129.6
20	*Y* = 1.126*X* + 1.721	0.9997	11.30–180.8
21	*Y* = 1.141*X* + 2.307	0.9996	11.75–188.0
22	*Y* = 1.174*X* + 1.252	0.9994	9.98–159.9
23	*Y* = 1.153*X* + 1.763	0.9995	18.51–295.8

**Table 2 molecules-26-04505-t002:** The Stability of 23 analytes from *S. officinalis*.

Compound	RE (%)				
	2 h	4 h	6 h	8 h	12 h
1	−0.5	−2.1	−1.1	1.6	2.2
2	−1.2	2.1	1.5	2.0	1.6
3	1.3	−2.1	−2.3	1.0	2.2
4	1.4	1.9	−0.8	1.3	2.1
5	2.1	1.7	2.2	1.5	−0.9
6	2.3	−2.7	1.9	2.0	−1.3
7	0.9	1.2	−1.5	−2.1	−2.8
8	1.9	−1.4	−1.8	−2.5	2.1
9	1.1	−1.7	1.5	2.9	−2.4
10	−0.9	1.3	−2.2	1.6	−1.9
11	−1.0	−1.2	1.6	2.0	2.1
12	1.1	−1.7	1.5	2.9	−2.4
13	−0.9	1.3	−2.2	1.6	−1.2
14	−1.1	1.9	1.7	−2.0	−2.8
15	1.4	−1.5	−1.0	2.2	1.4
16	−0.8	−1.3	2.1	1.4	−1.6
17	2.3	1.5	1.8	−2.3	1.2
18	1.0	−1.2	2.1	1.9	−0.9
19	0.8	1.1	−0.75	0.9	1.2
20	1.2	1.7	−1.4	1.0	2.0
21	−2.1	−1.1	−1.6	1.9	2.3
22	2.4	−1.0	0.2	0.4	−1.3
23	−2.0	1.8	1.1	1.4	−0.8

**Table 3 molecules-26-04505-t003:** Recoveries of 23 analytes in *S. officinalis*.

Compound	Recovery (%)	RE (%)
1	98.8	1.1
2	99.6	1.4
3	99.4	−1.1
4	99.1	1.5
5	98.9	−1.0
6	98.2	2.2
7	100.1	1.8
8	101.2	2.6
9	99.5	3.0
10	97.7	1.9
11	100.8	1.2
12	99.7	1.6
13	99.1	1.4
14	98.8	2.0
15	100.4	−1.3
16	101.4	1.0
17	100.6	3.0
18	98.9	2.1
19	99.5	2.0
20	99.2	2.4
21	101.1	−1.3
22	100.7	1.9
23	101.2	−2.3

**Table 4 molecules-26-04505-t004:** Thirty-two batches of *S. officinalis* from 24 provinces.

No.	Source	Collection Time	No.	Source	Collection Time
1	Anguo, Hebei	March 2016	17	Shanxi	March 2018
2	Shijiazhuang, Hebei	March 2016	18	Jizhou, Hubei	December 2016
3	Chengdu, Sichuan	April 2017	19	Kunming, Yunnan	October 2016
4	Mianyang, Sichuan	April 2017	20	Yulin, Guangxi	September 2016
5	Jilin	April 2017	21	Rizhao, Shandong	December 2017
6	Guiyang, Guizhou	March 2018	22	Qingdao, Shandong	December 2017
7	Zunyi, Guizhou	March 2018	23	Shenyang, Liaoning	March 2016
8	Hulunbeir, Inner Mongolia	June 2016	24	Jinzhou, Liaoning	April 2016
9	Chifeng, Inner Mongolia	June 2016	25	Harbin, Heilongjiang	June 2016
10	Shanxi	December 2016	26	Mudanjiang, Heilongjiang	October 2017
11	Lanzhou, Gansu	September 2018	27	Guangdong	November 2016
12	Tianshui, Gansu	September 2018	28	Jiangsu	May 2018
13	Bozhou, Anhui	December 2017	29	Hunan	December 2018
14	Pan’an, Zhejiang	May 2018	30	Qinghai	May 2017
15	Zhangshu, Jiangxi	April 2017	31	Chongqing	July 2017
16	Changzhou, Henan	May 2018	32	Tibet	November 2016

**Table 5 molecules-26-04505-t005:** Determination of 11 phenolic acids compounds of *S. officinalis* in 32 batches from 24 provinces (*n* = 3).

Source	Compound (%)											
	1	2	3	4	5	6	7	8	9	10	11	Total
DY01	3.72	0.45	5.94	0.33	0.07	0.55	0.18	2.93	0.24	0.85	1.91	17.17
DY02	3.45	0.41	5.86	0.32	0.05	0.61	0.24	3.15	0.12	0.97	2.07	17.25
DY03	1.29	0.24	3.75	0.16	0.10	0.25	0.26	2.87	ND	0.64	1.39	10.95
DY04	1.57	0.24	2.73	0.25	0.05	0.26	0.30	1.87	0.21	0.40	1.51	9.39
DY05	1.91	0.27	2.73	0.24	0.09	0.28	0.32	3.02	0.19	0.65	1.27	10.97
DY06	1.73	0.26	2.64	0.27	0.09	0.34	0.24	2.21	0.16	0.41	1.70	10.05
DY07	1.83	0.35	3.06	0.31	0.08	0.25	0.21	1.85	0.14	0.62	1.58	10.28
DY08	2.07	0.25	3.93	0.24	0.06	0.35	0.28	2.69	0.25	0.31	1.68	12.11
DY09	2.40	0.30	3.97	0.19	0.06	0.24	0.28	2.69	0.16	0.55	1.43	12.27
DY10	1.58	0.24	2.89	ND	0.10	0.27	0.10	1.59	0.11	0.47	1.47	8.82
DY11	2.36	0.30	3.28	0.15	0.10	0.25	0.31	2.33	0.23	0.70	1.79	11.80
DY12	2.88	0.39	3.42	0.33	ND	0.39	0.35	1.76	0.23	0.59	1.54	11.88
DY13	2.80	0.33	3.87	0.14	0.10	0.35	0.27	3.04	0.18	0.56	1.69	13.33
DY14	1.64	0.24	3.58	0.34	0.10	0.33	0.33	1.57	0.20	0.43	1.62	10.38
DY15	2.62	0.40	2.68	0.29	0.06	0.29	0.11	1.57	0.16	0.60	1.54	10.32
DY16	1.31	0.32	4.49	0.15	0.10	0.32	0.33	2.91	0.20	0.33	1.80	12.26
DY17	1.11	0.37	4.55	0.13	0.08	0.32	0.13	2.87	0.20	0.32	1.74	11.82
DY18	2.80	0.38	4.91	0.28	0.06	0.34	0.13	1.98	0.21	0.66	1.20	12.95
DY19	1.59	0.28	4.30	0.16	0.05	0.35	0.23	2.49	0.24	0.59	1.31	11.59
DY20	2.51	0.39	3.69	0.21	0.05	0.37	0.18	1.99	0.15	0.61	1.55	11.70
DY21	1.07	0.31	4.93	0.25	0.09	0.36	0.31	2.35	0.23	0.30	1.64	11.84
DY22	1.57	0.34	3.30	0.19	0.06	0.26	0.24	2.58	0.14	0.34	1.39	10.41
DY23	1.74	0.26	4.43	ND	0.08	0.20	0.30	2.54	0.15	0.57	1.30	11.57
DY24	2.97	0.31	3.08	0.23	0.05	0.22	ND	3.07	0.13	0.46	1.43	11.95
DY25	1.20	0.36	4.58	0.23	ND	0.21	0.23	1.70	0.16	0.31	1.50	10.48
DY26	2.70	0.28	3.41	0.16	0.08	0.35	0.33	3.08	0.17	0.35	1.42	12.33
DY27	2.01	0.34	3.46	0.23	ND	0.24	0.35	2.29	ND	0.62	1.57	11.11
DY28	2.71	0.24	4.08	0.32	0.05	0.31	0.30	2.67	0.13	0.61	1.55	12.97
DY29	1.83	0.35	2.91	0.11	0.09	0.29	0.16	2.79	0.25	0.35	1.61	10.74
DY30	2.72	0.39	3.56	0.10	ND	0.26	0.35	2.19	0.13	0.31	1.40	11.41
DY31	1.39	0.27	3.54	0.20	0.08	0.23	0.32	2.78	0.16	0.62	1.48	11.07
DY32	2.22	0.33	4.03	0.11	ND	0.31	0.23	2.88	0.18	0.60	1.52	12.41

ND = Not Detected.

**Table 6 molecules-26-04505-t006:** Determination of 12 triterpenes compounds of *S. officinalis* in 32 batches from 24 provinces (*n* = 3).

Source	Compound (%)												Total
	12	13	14	15	16	17	18	19	20	21	22	23	
DY01	5.93	0.29	0.30	1.03	0.30	0.75	2.07	0.29	0.65	2.23	0.10	3.18	17.12
DY02	5.88	0.37	0.21	1.07	0.25	0.77	2.84	0.34	0.77	2.39	0.23	2.16	17.28
DY03	3.18	0.43	0.28	0.87	0.41	0.65	1.97	0.38	0.44	1.21	0.13	2.14	12.09
DY04	4.65	0.29	0.59	0.91	0.18	0.92	1.57	0.30	0.61	1.31	0.12	1.81	13.26
DY05	2.35	0.23	0.61	0.47	0.15	0.85	1.42	0.34	0.80	1.58	0.14	2.28	11.22
DY06	4.27	0.16	0.46	0.46	0.35	0.69	1.89	0.16	0.36	1.61	0.17	1.88	12.46
DY07	2.29	0.49	0.30	0.71	ND	0.54	1.43	0.25	0.37	1.10	0.15	1.32	8.95
DY08	3.60	0.17	0.67	0.41	0.20	0.94	1.69	0.44	0.36	1.11	0.15	1.00	10.74
DY09	5.24	0.43	0.67	0.99	0.26	0.78	1.59	0.12	0.50	1.24	0.18	2.03	14.03
DY10	3.15	0.33	0.78	0.47	0.21	0.93	1.42	0.22	0.20	1.62	0.16	1.49	10.98
DY11	3.53	0.42	0.22	0.74	ND	0.66	1.32	0.24	0.53	1.19	0.17	1.00	10.02
DY12	3.29	0.55	0.34	0.65	0.31	0.72	1.93	0.19	0.79	1.47	0.13	2.03	12.40
DY13	4.44	0.22	0.60	0.40	0.17	0.62	1.29	0.30	0.36	1.12	ND	1.69	11.21
DY14	4.69	0.32	0.15	0.82	0.38	0.78	1.33	0.26	0.32	1.11	0.13	1.75	12.04
DY15	5.39	0.31	0.19	0.56	0.26	0.79	2.06	0.08	0.24	1.65	0.15	2.40	14.08
DY16	3.58	ND	0.58	0.53	0.44	0.92	1.83	0.34	0.41	1.80	0.16	2.38	12.97
DY17	2.85	0.54	0.43	0.55	0.34	0.81	1.31	0.47	0.45	2.06	0.15	1.04	11.00
DY18	3.36	0.40	0.31	0.68	ND	0.62	2.01	0.27	0.23	1.86	0.16	1.71	11.61
DY19	3.01	0.36	0.51	0.42	0.32	0.76	2.06	ND	0.54	1.10	0.18	1.80	11.06
DY20	5.19	0.58	ND	0.60	0.15	0.73	1.25	0.31	0.72	1.01	0.12	1.02	11.68
DY21	3.40	0.26	0.40	0.87	0.30	0.81	1.49	0.46	0.49	1.49	0.12	2.45	12.54
DY22	4.36	0.55	0.54	0.65	0.24	0.58	2.03	0.40	0.35	2.05	0.13	1.33	13.21
DY23	5.30	0.54	0.33	0.47	0.18	0.83	1.67	0.43	0.68	1.12	0.12	1.42	13.09
DY24	3.05	0.16	ND	0.86	0.19	0.95	1.21	0.29	0.53	1.14	0.12	2.08	10.58
DY25	4.50	0.52	0.47	0.43	0.50	0.69	1.68	0.24	0.44	1.85	0.18	2.17	13.67
DY26	4.46	0.24	0.78	0.58	0.38	1.00	1.33	0.22	0.66	1.60	0.12	1.49	12.86
DY27	3.64	ND	0.41	0.51	0.41	0.63	1.31	0.27	0.60	2.07	0.14	1.79	11.78
DY28	2.65	0.60	0.69	0.94	0.19	0.77	1.93	0.48	0.59	1.72	0.18	1.71	12.45
DY29	5.07	0.35	0.62	0.84	ND	0.69	1.80	0.05	0.40	1.17	0.13	2.02	13.14
DY30	2.87	0.33	0.38	0.57	0.17	0.70	1.75	0.38	0.55	1.57	0.17	1.16	10.60
DY31	3.03	0.29	0.30	0.99	0.30	0.96	1.89	ND	0.21	1.54	0.17	1.71	11.39
DY32	4.59	0.58	0.78	0.53	0.20	0.79	1.86	0.35	0.23	1.48	ND	1.21	12.60

ND = Not Detected.

**Table 7 molecules-26-04505-t007:** Data for determination of 23 compounds in unprocessed *S. officinalis* and processed *S. officinalis* from 3 provinces (*n* = 3).

Compound	Source					
	Heilongjiang		Henan		Anhui	
	Unprocessed	Processed	Unprocessed	Processed	Unprocessed	Processed
1	1.19	0.75 **	1.32	1.03 **	2.82	1.35 **
2	0.38	0.42	0.33	0.41	0.34	0.44
3	4.56	0.22 **	4.48	0.19 **	3.85	0.27 **
4	0.23	1.56 **	0.15	2.49 **	0.13	2.25 **
5	ND	ND	0.11	ND **	0.11	ND **
6	0.21	1.10 **	0.33	0.42	0.34	1.63 **
7	0.23	4.21 **	0.31	2.10 *	0.26	1.32 **
8	1.71	0.84 ^*^	2.89	0.76 **	3.02	1.27 **
9	0.16	0.04 **	0.21	0.04 **	0.18	0.04 **
10	0.32	8.83 **	0.34	3.46 **	0.57	7.54 **
11	1.52	0.01 **	1.82	0.01 **	1.67	0.02 **
12	4.49	0.44 **	3.56	2.89	4.42	1.30 **
13	0.53	3.58 **	ND	0.82 *	0.21	0.47 *
14	0.48	0.77 **	0.56	0.12 **	0.58	0.56
15	0.42	ND **	0.51	ND **	0.39	ND **
16	0.49	ND ^**^	0.42	ND **	0.17	ND **
17	0.66	1.32 **	0.92	0.96	0.62	0.13 **
18	1.70	0.76 **	1.80	0.46 *	1.28	0.22 **
19	0.23	1.20 **	0.32	4.77 *	0.31	0.26 **
20	0.45	0.55	0.40	0.37	0.36	0.22
21	1.79	0.14 **	1.82	0.09 **	1.11	0.03 **
22	0.17	0.79 *	0.17	0.91 **	ND	ND
23	2.15	1.59 *	2.40	0.86 **	1.68	0.48 **

ND = Not Detected; * *p* < 0.05, ** *p* < 0.01, compared with that of the unprocessed group.

**Table 8 molecules-26-04505-t008:** ^13^C NMR data (Pyridine-*d_5_*, 100 MHz) of compound 13–18 and 20–23.

No.	13	14	15	16	17	18	20	21	22	23
1	48.5	38.6	39.9	39.4	39.1	39	43.2	81.0	39.0	47.9
2	69.1	26.4	27.2	27.2	28.1	32.6	66.5	71.6	27.2	68.3
3	84.4	88.5	89.2	89.2	78.3	216.1	79.6	79.8	78.2	83.6
4	39.0	40.3	40.1	40.1	39.4	47.2	39.1	38.6	39.4	39.8
5	56.5	55.7	56.6	56.5	55.9	55.3	49.1	48.6	55.9	55.7
6	19.5	18.5	19.2	19.0	19.0	19.7	18.8	18.9	19.0	18.6
7	34.0	33.3	35.8	33.9	33.6	34.2	33.7	33.9	33.6	33.2
8	41.1	39.3	40.0	40.3	40.6	40.2	40.3	41.3	40.4	39.7
9	48.4	47.5	48.8	48.6	47.8	46.6	48.3	48.8	47.8	48.1
10	40.3	36.8	37.4	37.6	37.4	36.7	39.1	43.9	37.4	38.5
11	24.7	23.8	24.0	24.4	24.1	23.8	24.2	26.6	24.1	23.6
12	128.8	128.2	127.0	129.1	128.5	127.6	128.8	129.6	128.1	122.2
13	139.8	139.0	139.2	138.0	139.3	139.7	137.9	138.9	140.0	144.6
14	42.6	41.9	45.4	43.3	42.1	42.5	43.2	42.2	42.1	42
15	29.7	29.0	29.6	29.6	29.3	28.7	29.4	29.5	29.4	28.2
16	26.6	25.9	27.3	26.3	26.2	26.3	26.0	28.1	26.4	23.5
17	49.1	48.4	50.4	50.3	48.7	47.2	50.2	47.9	48.3	46.4
18	54.9	54.2	123.7	52.7	54.5	54.3	52.6	54.6	54.7	42
19	73.1	72.4	134.3	153.9	72.7	72.3	153.7	72.8	72.7	46.4
20	42.6	41.9	35.1	38.0	42.1	42.4	37.9	42.5	42.4	30.9
21	27.2	26.4	31.5	31.1	26.7	35.7	31.0	27.2	24.7	34.2
22	38.2	37.5	35.6	37.6	37.7	74.8	37.4	38.6	38.5	33.1
23	29.8	28.0	28.8	28.7	28.8	26.7	29.7	29.5	28.8	29.3
24	17.2	16.7	16.8	17.4	16.5	21.4	22.4	22.4	16.5	17.5
25	17.5	15.4	16.6	16.2	15.7	14.7	17.7	13.2	15.6	16.6
26	18.0	17.2	18.6	17.8	17.5	16.7	17.2	17.7	16.8	17.2
27	25.0	24.3	22.6	26.7	24.6	24.6	26.6	24.8	24.7	25.9
28	177.4	176.7	175.1	176.5	177.0	179.6	176.5	180.9	180.7	180
29	27.5	26.8	20.0	110.9	27.1	19.1	110.8	27.2	27.2	33.2
30	18.1	16.4	18.9	19.9	16.7	16.4	19.7	16.9	17.3	23.7
1′		107.2	108.0	108.0						
2′		72.7	73.4	73.4						
3′		74.7	74.7	75.1						
4′		69.3	70.0	70.0						
5′		66.5	67.2	67.2						
1″	96.3	95.6	96.3	96.4	95.9		96.2			
2″	74.6	73.8	75.1	74.6	74.1		74.3			
3″	79.7	79.0	79.6	79.4	79.3		79.1			
4″	71.7	71.0	71.7	71.6	71.2		71.5			
5″	79.4	78.7	79.6	79.8	78.9		79.6			
6″	62.8	62.1	62.8	62.8	62.4		62.6			

## Data Availability

Not applicable.

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
