# Peer review of "Determination of Triterpenoids and Phenolic Acids from *Sanguisorba officinalis* L. by HPLC-ELSD and Its Application"

_molecules, 2021, doi:10.3390/molecules26154505_

Round 1
Reviewer 1 Report
As far as my comments as reviewer of the original version are concerned, the authors have revised the manuscript in a reasonable way.
Unfortunately, I am sorry to say that all in all the revised manuscript looks worse than the original version. The parts containing new text (most of them probably a response to the comments of the other reviewer) contain a lot of mistakes with the English.
Furthermore, it is difficult to see how the new lines 62 to 80 fit to the rest of the text. In my opinion, some of the references given here are not relevant in this context. If the authors intended to give arguments for the use of ELSD, a few much shorter sentences would be sufficient (but I do not know what the reasons had been for including this new paragraph).
Lines 225 to 236: I cannot see how the LC/MS/MS data would contribute in a reasonable way to the topic reported in this manuscript. If these data are taken from existing papers, it would be sufficient just to mention those papers (but I cannot see how one would need these data at all in the present context).
Lines 246 to 317: As far as I can see, this part contains 1H NMR data for most of the analytes , but not for all of them (?). It also seems that some 13C NMR data are included in lines 246 to 317, whereas others are listed in Table 8. This looks a bit confusing.
What is meant by "RE" in line 361 ?
Author Response
Point 1: As far as my comments as reviewer of the original version are concerned, the authors have revised the manuscript in a reasonable way.
Unfortunately, I am sorry to say that all in all the revised manuscript looks worse than the original version. The parts containing new text (most of them probably a response to the comments of the other reviewer) contain a lot of mistakes with the English.
Response 1: Please provide your response for Point 1. The mistakes with English expression have been corrected: page 2, lines 62 to 67; page2, lines 69 to 70; page12, line 218, line 237 and line 239; page 16, line 392, line394, line 407, and line 408.
Point 2: Furthermore, it is difficult to see how the new lines 62 to 80 fit to the rest of the text. In my opinion, some of the references given here are not relevant in this context. If the authors intended to give arguments for the use of ELSD, a few much shorter sentences would be sufficient (but I do not know what the reasons had been for including this new paragraph).
Response 2: Please provide your response for Point 2. Lines 62 to 80 is not a new paragraph, but an existing paragraph in the original version, and it is also a revised content based on the comments of reviewer 2 of the original version. At the same time, the purpose of this paragraph is to show that ELSD could eliminate the problems of traditional method in content determination.
Point 3: Lines 225 to 236: I cannot see how the LC/MS/MS data would contribute in a reasonable way to the topic reported in this manuscript. If these data are taken from existing papers, it would be sufficient just to mention those papers (but I cannot see how one would need these data at all in the present context).
Response 3: Please provide your response for Point 3. For the LC/MS/MS data provided in lines 225 to 236, we provided the data in accordance the reviewer’s comments of the original version. The reviewer 3’s comment mentioned that “I suggest that the authors should include at least LC/MS/MS data in discussion and link with another secondary metabolites isolated previously."
Point 4: Lines 246 to 317: As far as I can see, this part contains 1H NMR data for most of the analytes, but not for all of them (?). It also seems that some 13C NMR data are included in lines 246 to 317, whereas others are listed in Table 8. This looks a bit confusing.
Response 4: Please provide your response for Point 4. For the NMR data provided in lines 246 to 317, we provided the data in accordance the reviewer’s comments of the original version. The comments of reviewer 2 of the original version mentioned that “NMR and MS data about the 23 isolated compounds should be reported, as they are used as reference standards.”
Point 5: What is meant by "RE" in line 361 ?
Response 5: Please provide your response for Point 5. RE means relative error, and it's the ratio of absolute error and true value. In the manuscript we used the value of RE to show the stability of the method. It is the actual difference between stored at room temperature at 2, 4, 6, 8, 12 h and stored at room temperature at 0 h.
Reviewer 2 Report
Authors addressed my comments and modified their work accordingly. The paper is suitable for publication in the actual form.
Author Response
Point 1: Authors addressed my comments and modified their work accordingly. The paper is suitable for publication in the actual form.
Response 1: Please provide your response for Point 1. Thank you very much for your comments and suggestions for our manuscript.
Reviewer 3 Report
Dear Authors
the revised version of the MS 1251926 may be accepted in its present form.
Author Response
Point 1: the revised version of the MS 1251926 may be accepted in its present form.
Response 1: Please provide your response for Point 1. Thank you very much for your comments and suggestions for our manuscript.
Reviewer 4 Report
It is always interesting separation of components in one run. The problem is well presented and dissolved. I suggest to give short information in figures. It is always better and more clear for readers.
Author Response
Point 1: It is always interesting separation of components in one run. The problem is well presented and dissolved. I suggest to give short information in figures. It is always better and more clear for readers.
Response 1: Please provide your response for Point 1. The short information for the figure of separation is provided in section 2.1. page4, lines 101 to 114. "According to the selected optimal chromatographic conditions, the retention times of 23 analytes are: gallic acid (10.131 min), sanguiin H-4 (17.347 min), catechin (23.620 min), methyl-6-O-galloyl-β-D-glucopyranoside (26.492 min), caffeic acid (31.465 min), syringic acid (32.702 min), ethyl gallate (36.068 min), 3,4-dihydroxy-5-methoxybenzyl ester (39.079 min), ferulic acid (43.685 min), ellagic acid (51.505 min), 3,3-dihydroxy ellagic acid (57.338 min), 2-oxo-3β,19α-dihydroxy-olean-12-en-28-oic acid β-D-glucopyranosyl ester (66.723 min), rosamultin (73.744 min), ziyuglycoside Ⅰ (78.160 min), 3β-[(α-L-arabinopyranosyl)oxy]-urs-12,18(19)-dien-28-oic acid β-D-glucopyranosyl ester (88.975 min), 3β-[(α-L-arabinopyranosyl)oxy]-urs-12,19(29)-dien-28-oic acid β-D-glucopyranosyl ester (94.293 min), 3β,19α-dihydroxyurs-12-en-28-oic-acid 28-β-D-glucopyranosyl ester (98.958 min), pomonic acid (101.456 min), euscaphic acid (103.725 min), 2α, 3α-duhydroxyursa-12,19(29)-dien-28-oic acid β-D-glucopyranosyl ester (105.710 min), 1β-hydroxyeuscaphic acid (108.168 min), 3β,19α-dihydroxy-12-ursen-28-acid (111.653 min) and 2α-hydroxyoleanolic acid (121.715 min)."
Reviewer 5 Report
The manuscript entitled "Determination of triterpenoids and phenolic acids from Sanguisorba officinalis L. by HPLC-ELSD and its application" reports interesting results in the field of the analytical chemistry of complex plant extracts.
The reviewer addresses to the authors the following remarks that should be considered before publication.
As a general remark, substantial parts of the text are highlighted in red, as if the manuscript were still in a preparatory state and not fully ready for submission.
page 1, line 41. catechin is not an acid but a polyphenol
page 2, line 50. Compound 23 is 2α-hydroxyoleanolic acid, not 2α-hydroxydenolic acid. As well in page 4, line 115.
page 3, Figure 1. There is a stray "G-" besides structure 4 and an unidentified methyl glucoside between structures 3 and 5.
page 4, lines 98-102. Detailed HPLC conditions are not useful here and were also repeated in the Methods section.
By the way, the chromatograms are really nice.
page 4, line 117. Use of "double logarithmic transformation" should be at least shortly discussed. The R2 correlation coefficient indicates a good linearity between the logarithms of the areas (A) and the concentrations (c). Proportionality of A and c would result is an equation like ln(A) = 1.ln(c) + b. Having ln(A) = a.ln(c) + b indicates a non-linearity if a different from 1. Moreover, the values of a, ranging from 0.8546 to 1.174 in Table 1, is not simply an instrument-dependent factor as it also depends on compound. This requires an explanation.
page 5, line 125. The distinction between Precision, Repeatability, and Stability is not clear, even after the reading of section 4.5.2. What is RE?
page 8, Table 5 and Table 6. What is the unit of these numbers? What do they report?
page 10, line 178. The concepts of "unprocessed" and "processing" appears here for the first time. Please replace "processing" by "processed" at all the places this makes sense. Please explain what is a processed sample here. Why is there a need for a processing? Is this a requirement for the therapeutic activity to take place?
page 11, line 196. "Linear of the method over a wide concentration range..." Is this direct linearity or linearity of the logarithms of areas with logarithms of concentrations?
page 12, line 213. A special character was left at the place of a Greek letter, alpha or beta.
page 12, lines 226-236. These data would be better placed in section 4.1.
page 14, line 314. This one is 2α-hydroxyoleanolic acid.
page 15, line 330. Unit is missing.
page 16, line 376. Yes, sample processing is described here. To which extent is this "burning" procedure reproducible? It seems that the process relies on some human expertise to determine how black is black and brown is brown (light brown, dark brown?).
What is black? Carbon? Is the processed material still soluble?
Author Response
Point 1: page 1, line 41. catechin is not an acid but a polyphenol
Response 1: Please provide your response for Point 1. In the revised version, we change phenolic acids to polyphenol in page 1, line40.
Point 2: page 2, line 50. Compound 23 is 2α-hydroxyoleanolic acid, not 2α-hydroxydenolic acid. As well in page 4, line 115.
Response 2: Please provide your response for Point 2. Compound 23 in page 2, line 50, and page 4, line 115 have been revised to 2α-hydroxyoleanolic acid in page 2, line 50, and page 4, line 126.
Point 3: page 3, Figure 1. There is a stray "G-" besides structure 4 and an unidentified methyl glucoside between structures 3 and 5.
Response 3: Please provide your response for Point 3. Figure 1 in page 3, structure 4 has revised in the new Figure 1 in page 3.
Point 4: page 4, lines 98-102. Detailed HPLC conditions are not useful here and were also repeated in the Methods section.
By the way, the chromatograms are really nice.
Response 4: Please provide your response for Point 4. Detailed HPLC conditions in page 4, lines 98-102 has revised as follows: “Determined the samples and 23 analytes according to the selected optimal chromatographic conditions. The representative HPLC-ELSD chromatograms of test samples and 23 analytes are shown in Figure 2.” in page4, lines 98-100.
Point 5: page 4, line 117. Use of "double logarithmic transformation" should be at least shortly discussed. The R2 correlation coefficient indicates a good linearity between the logarithms of the areas (A) and the concentrations (c). Proportionality of A and c would result is an equation like ln(A) = 1.ln(c) + b. Having ln(A) = a.ln(c) + b indicates a non-linearity if a different from 1. Moreover, the values of a, ranging from 0.8546 to 1.174 in Table 1, is not simply an instrument-dependent factor as it also depends on compound. This requires an explanation.
Response 5: Please provide your response for Point 5. Explanation of the use of "double logarithmic transformation" shown in page 5, line 129 to 135. “Linearity was evaluated by five different concentrations of standard solutions. For calibration, the standard solutions with five different concentrations were analyzed. A plot of peak area versus sample size by ELSD was not linear, but the plot of peak area versus sample size in double logarithm was linear, which conformed to the mechanism of the ELSD. Their regression equations were calculated in the form of Y =a X + b, where Y and X are the logarithms of the peak area and concentration, respectively. As shown in Table 1, compounds of different structures also show different slopes.”
Point 6: page 5, line 125. The distinction between Precision, Repeatability, and Stability is not clear, even after the reading of section 4.5.2. What is RE?
Response 6: Please provide your response for Point 6. The distinction between Precision, Repeatability, and Stability in the article is as follows: Precision refers to the degree of closeness between the results obtained from multiple injections of the same sample under the specified measurement conditions, and is used to evaluate the closeness of the results measured by the instrument when the same sample is injected. Repeatability was confirmed with six independent analytical sample prepared as preparation of standard method, which is used to evaluate whether the prepared samples are parallel. Stability is to assess whether the content of the sample will change at room temperature. RE means relative error, and it’s the ratio of absolute error and true value. In the manuscript we used the value of RE to show the stability of the method.
Point 7: page 8, Table 5 and Table 6. What is the unit of these numbers? What do they report?
Response 7: Please provide your response for Point 7. The unit of numbers in Table 5 and Table 6 is “%”. The numbers in Table 5 and Table 6 report the content of 23 compounds in the sample in different batches.
Point 8: page 10, line 178. The concepts of "unprocessed" and "processing" appears here for the first time. Please replace "processing" by "processed" at all the places this makes sense. Please explain what is a processed sample here. Why is there a need for a processing? Is this a requirement for the therapeutic activity to take place?
Response 8: Please provide your response for Point 8. What is a processed sample had shown in section 4.7. This method was carried out in accordance with the Chinese Pharmacopoeia (2020 edition).
We have replaced all “processing” with “processed”.
Based on the traditional Chinese theory, the unprocessed S. officinalis can cool the blood and detoxify, while the processed S. officinalis can stop bleeding. The unprocessed S. officinalis and the processed S. officinalis show different pharmacological effects in clinical practice. In order to provide ideas for the future indication of pharmacological effects of the chemical composition and the changes in the efficacy and the effective substances before and after processing in S. officinalis, we compared the unprocessed and processed S. officinalis from several origins and studied the changes in the content of these compounds.
Point 9: page 11, line 196. "Linear of the method over a wide concentration range..." Is this direct linearity or linearity of the logarithms of areas with logarithms of concentrations?
Response 9: Please provide your response for Point 9. As the explanation in page 5, line 129 to 135. It’s the linearity between the logarithm of the concentration and the logarithm of the peak area.
Point 10: page 12, line 213. A special character was left at the place of a Greek letter, alpha or beta.
Response 10: Please provide your response for Point 10. The place left in page 12, line 213 is beta, and we have corrected it in page 12, line 230.
Point 11: page 12, lines 226-236. These data would be better placed in section 4.1.
Response 11: Please provide your response for Point 11. The data in page 12, lines 226-236 has been placed in section 4.1. in the revised version in page 14, lines 323-334.
Point 12: page 14, line 314. This one is 2α-hydroxyoleanolic acid.
Response 12: Please provide your response for Point 12. Compound 23 in page 14, line 314 has been revised to 2α-hydroxyoleanolic acid in page 13, line 319.
Point 13: page 15, line 330. Unit is missing.
Response 13: Please provide your response for Point 13. The unit in page 15, line 330 is “μg/mL”, has corrected in page 15, line 347 of the revised version.
Point 14: page 16, line 376. Yes, sample processing is described here. To which extent is this "burning" procedure reproducible? It seems that the process relies on some human expertise to determine how black is black and brown is brown (light brown, dark brown?).
What is black? Carbon? Is the processed material still soluble?
Response 14: Please provide your response for Point 14. In order to make the processing procedure reproducible, after our early exploration of the processing methods, we specified the same temperature (120 ℃) and processing time (15 min) during the processing to ensure that the surface and inside colors of the processed S. officinalis are similar.
The processed S. officinalis still prepared according to the method in section 4.2. and it is made into a solution.
This manuscript is a resubmission of an earlier submission. The following is a list of the peer review reports and author responses from that submission.
Round 1
Reviewer 1 Report
In general, I cannot recommend a publication of this manuscript.
- The authors use some strange terms for the operating conditions of the ELSD, namely "drift tube temperature" and "atomizing tube temperature". Since the principles of an ELSD do not involve an atomization of the analyte, the term "atomizing tube temperature" is misleading and wrong. I also checked the manual for this detector (available on the Internet), and I could not find these two terms in the manual. Instead, there is a nebulizer temperature and an evaporator temperature.
- In the Conclusions the authors say "a rapid, sensitive and reliable quantitative method for simultaneous analysis of phenolic acids and triterpenoids from S. officinalis was developed". I do not agree that the method would be rapid. The authors use a 250 mm long column packed with 5 µm particles, and Figure 2 shows that the separation takes more than 2 hours (!). In my opinion, the use of a modern UHPLC column may lead to considerably shorter separation times.
- The authors have analyzed one specific plant material used as a traditional Chinese medicine (Sanguisorba officinalis L.), whereby the chemical constituents are known anyway (the authors say in lines 42 and 43 "The chemical constituents isolated from S. officinalis are mainly triterpenoids and glycosides [14], phenolics, and flavonoids [15,16]". It may be true that the Chinese Pharmacopoeia includes only gallic acid as quality control indicator, so that it may be justified to establish a quality control method that includes more of the constituents. However, the development of the HPLC method follows well established approaches and looks pretty much like routine work.
- Different batches of S. officinalis from different areas have been analyzed. The authors say correctly "The content of triterpenoids and phenolic acids in S. officinalis are affected by many factors such as region, environment and harvest time, so there are some differences
in the contents". It is unclear what a reader can learn from the results of these samples that seem to have been selected randomly. - The authors investigate the difference between unprocessed and processed material. The processing step is described as "S. officinalis of processing group was fried in the 120 ℃ iron pan for 15 min". For a reader it is unclear why these processing conditions have been selected. It is not surprising that the concentrations of the analytes change after this processing step, but it is unclear what sort of scientific conclusions can be drawn from these results.
- All in all, I believe that this manuscript describes one specific case study of a traditional Chinese medicine, but I do not believe that it contains fundamental scientific novelty that would be of interest for a wider range of readers.
Reviewer 2 Report
The work by Sun and coll. describes an innovative LC-ELSD method for the analysis of the characteristic compounds of Sanguisorba officinalis, a medicinal plant used in the traditional Chinese medicine. The aim is to monitor different classes of compounds (i.e. triterpenes, catechin and phenolic acids) using a single chromatographic run, employing an “universal” detector that allows to collect signals from all the classes of compounds (also those that are difficultly analyzed using a DAD). Although the idea is valuable, I think that the applicability of this method is low, due mainly to two reasons: 1) the length of the chromatographic run: 2 h for a single analysis are too much, especially if the method was developed for a routinary quality control. The analysis of large batches would be really time consuming and scarcely adoptable for a QC lab; 2) ELSD detector and lack of reference standards: the assignment of peaks in LC-ELSD requires the co-injection with reference compounds. However, the authors isolated these constituents directly from dried S. officinalis, but many of these compounds are difficult to find on the market. This hampers the applicability of the method by other labs that could not isolate the compounds by themselves.
Overall, I do not recommend this work for publication in Molecules. However, if the editor will choose to consider it for publication, I suggest the authors to address some major issues that emerged during the revision process.
Whole manuscript
- Check writing language. There are several spelling and grammar errors throughout the manuscript.
Introduction
- Lines 38-39: “….anti-oxidation [4], anti-inflammation [5], antivirus [6], antibiosis [7,8] and anti-tumor [9].” Should be changed with “antioxidant, anti-inflammatory, antiviral, antimicrobial, and antitumor”.
- Line 42: “glycosides” means “triterpene glycosides”?
- Line 44: catechin is not a phenolic acid
- Figure 4 should be improved, especially the upper part regarding phenolic acids and catechin
- Lines 66-82: repetitive parts, you should condense this section
- Abbreviations should be introduced at their first appearance in the manuscript, and then they should be used instead of the full terms (e.g. HPLC)
Results
- The method could not be defined as “rapid”, as the length is more than 2 h. This means that its application in routine QC analysis would be difficult, especially for the analysis of large batches of samples.
- The determination of inter-day precision is completely lacking
- Line 140: What “RE” means?
- Line 153: “All medicinal materials are purchased in the local Chinese medicine market.” The sentence needs more specifications.
Discussion
- This part does not report any comment about method development and validation (that should be the aim of this work), as well as any comparison with previously published literature. Only a description of data already reported in the tables and figures above is reported, again with no contextualization or comparison with available literature. This chapter should be drastically improved.
Materials and Methods
- NMR and MS data about the 23 isolated compounds should be reported, as they are used as reference standards.
- Lines 258-261: it is not clear how you calculated the recovery
Conclusion
- The method could not be defined as “rapid”, as the length is more than 2 h. This means that its application in routine QC analysis would be difficult, especially for the analysis of large batches.
Reviewer 3 Report
Dear Authors
I suggest that the authors should include at least LC/MS/MS data in discussion and link with another secondary metabolites isolated previously. This is important for future studies and show that HPLC/ELSD is a powerful technique and avoid dereplication. Despite the relevance of some data from this study, the results presented here are enough from the chemical point of view for publishing it.